# Relative Risks of Adverse Perinatal Outcomes in Three Australian Communities Exposed to Per- and Polyfluoroalkyl Substances: Data Linkage Study

**DOI:** 10.3390/ijerph20196886

**Published:** 2023-10-05

**Authors:** Hsei Di Law, Deborah A. Randall, Bruce K. Armstrong, Catherine D’este, Nina Lazarevic, Rose Hosking, Kayla S. Smurthwaite, Susan M. Trevenar, Robyn M. Lucas, Archie C. A. Clements, Martyn D. Kirk, Rosemary J. Korda

**Affiliations:** 1National Centre for Epidemiology and Population Health, Australian National University, Canberra, ACT 2601, Australia; 2Women and Babies Research, Northern Clinical School, The University of Sydney, St Leonards, NSW 2050, Australia; 3School of Population and Global Health, The University of Western Australia, Perth, WA 6009, Australia; 4Office of Vice Chancellor, University of Plymouth, Plymouth PL4 8AA, UK

**Keywords:** PFAS, perinatal, pregnancy, mothers, birth outcomes, firefighting foams

## Abstract

Introduction: Firefighting foams containing per- and polyfluoroalkyl substances (PFAS) have caused environmental contamination in several Australian residential areas, including Katherine in the Northern Territory (NT), Oakey in Queensland (Qld), and Williamtown in New South Wales (NSW). We examined whether the risks of adverse perinatal outcomes were higher in mothers living in these exposure areas than in selected comparison areas without known contamination. Methods: We linked residential addresses in exposure areas to addresses collected in the jurisdictional Perinatal Data Collections of the NT (1986–2017), Qld (2007–2018), and NSW (1994–2018) to select all pregnancies from mothers who gave birth while living in these areas. We also identified one comparison group for each exposure area by selecting pregnancies where the maternal address was in selected comparison areas. We examined 12 binary perinatal outcomes and three growth measurements. For each exposure area, we estimated relative risks (RRs) of adverse outcomes and differences in means of growth measures, adjusting for sociodemographic characteristics and other potential confounders. Results: We included 16,970 pregnancies from the NT, 4654 from Qld, and 7475 from NSW. We observed elevated risks of stillbirth in Oakey (RR = 2.59, 95% confidence interval (CI) 1.25 to 5.39) and of postpartum haemorrhage (RR = 1.94, 95% CI 1.13 to 3.33) and pregnancy-induced hypertension (RR = 1.88, 95% CI 1.30 to 2.73) in Williamtown. The risks of other perinatal outcomes were not materially different from those in the relevant comparison areas or were uncertain due to small numbers of events. Conclusions: There was limited evidence for increased risks of adverse perinatal outcomes in mothers living in areas with PFAS contamination from firefighting foams. We found higher risks of some outcomes in individual areas, but these were not consistent across all areas under study and could have been due to chance, bias, or confounding.

## 1. Introduction

Per- and polyfluoroalkyl substances (PFAS) are man-made chemicals widely used for household and industrial purposes since the 1950s. The movement of PFAS through water and land has led to environmental contamination globally [1,2,3]. These chemicals are resistant to environmental degradation and are easily absorbed, distributed, and retained in the human body [4,5].

In Australia, PFAS contamination has occurred in residential areas surrounding military bases where firefighting foams containing predominantly perfluorooctane sulfonic acid (PFOS) and perfluorohexane sulfonic acid (PFHxS) were used. Although these particular foams have been phased out since the 2000s, PFAS remain detectable in water sources and land near military bases [6,7,8]. Affected residential areas in Australia include Katherine in the Northern Territory (NT), Oakey in Queensland (Qld), and Williamtown in New South Wales (NSW) (hereafter referred to as exposure areas).

Measurements made from 2016 to 2019 in these areas provide some context for the levels of exposure, although serum collected at one point in time do not reflect long-term cumulative exposure or changes over time. The geometric means of serum PFAS ranged from 4.9 to 6.6 ng/mL for PFOS, 2.9 to 3.7 ng/mL for PFHxS, and 1.3 to 1.8 ng/mL for perfluorooctanoic acid (PFOA). The levels of PFOS and PFHxS in these areas were higher than in people from selected comparison communities without known contamination, while the levels of PFOA were similar [9]. These serum concentrations are comparable to those reported in three US communities [10,11,12] but below those of residents in Ronneby, Sweden [13], also affected by firefighting foams.

Due to the ability of PFAS to cross the human placenta [14,15], some epidemiological investigations have focused on maternal and infant health outcomes. These studies have involved three types of population: the general population (background exposure), communities living in areas with documented contamination of the local environment or drinking water supply (community exposure, as in the present study), and, to a lesser extent, workers exposed in plants or personnel using fighting foams (occupational exposure).

Most epidemiological studies on PFAS have focused on PFOA and/or PFOS. Fewer data are available for other types of PFAS. Based on these studies, the Agency for Toxic Substances and Disease Registry (ATSDR) and multiple reviews have supported associations between PFOA and PFOS and small reductions in birthweight [16,17,18,19]. Additionally, largely based on studies of community exposure in the mid-Ohio Valley region of the US from the C8 Health Project [20], the ATSDR and a recent authoritative review cited ‘suggested’ [17] and ‘probable’ [16] links, between PFOA/PFOS and pregnancy-induced hypertension and preeclampsia, respectively.

The aim of this study was to examine whether the risks of adverse perinatal outcomes were higher among mothers who gave birth at the time of living in a PFAS-exposure area in Australia compared to those living in comparison areas without known contamination. This can help inform the community of the possible excess risk of living in these areas due to local PFAS contamination.

## 2. Methods

### 2.1. Data Sources and Study Population

Data were sourced from jurisdictional Perinatal Data Collections (PDC) of the NT, Qld and NSW. These collections contain perinatal records of each mother’s pregnancy and birth outcomes, as well as the mother’s address and other demographic information. All live births and stillbirths of at least 400 g birthweight or at least 20 weeks gestation are included in these data. In this study, NT perinatal data were available from 1986 to 2017, for Qld from 2007 to 2018, and for NSW from 1994 to 2018.

Exposed populations were defined as pregnancy records in the NT, Qld, or NSW PDC where the maternal residential address was in the three exposure areas of interest: Katherine, NT, Williamtown, NSW, and Oakey, Qld, within boundaries as defined by the Australian Department of Defence [21,22,23]. We extracted all street addresses that fell inside the boundaries of these exposure areas from the Australian Geocoded National Address File (G-NAF) [24].

The comparison populations were defined as pregnancy records in the NT, Qld, or NSW PDC where the maternal residential address was in any comparison area. Comparison areas were a selected list of postcodes chosen on the basis that they had similar sociodemographic profiles to the exposure areas, according to Australian Bureau of Statistics data. We chose as many postcodes as necessary to obtain comparison populations that were approximately four times that of the relevant exposed population. The comparison area postcodes for the three exposure areas were: for Katherine: 0800, 0828, 0829, 0835, 0836, 0837, 0838, 0840, 0841, 0845, 0846, 0880, 0886; for Oakey: 4311, 4371, 4372, 4373, 4610; and for Williamtown: 2334, 2335, 2864, 2865, 2866, 2867, 2477.

Linkages of maternal address data in the PDCs to extracted addresses from G-NAF were performed by third-party specialist data linkage organisations in the NT, Qld and NSW.

### 2.2. Variables

#### 2.2.1. Outcomes

We examined 15 outcomes: 12 binary outcomes and three growth measurements. Outcomes, either pre-defined in the datasets or based on International Classification of Diseases Australian Modification, 10th revision (ICD-10-AM) codes were: gestational diabetes (O24.4), pregnancy-induced hypertension (O13, O14), caesarean or assisted vaginal birth, emergency caesarean, postpartum haemorrhage (O72), preterm birth (<37 weeks of gestation), spontaneous preterm birth, small for gestational age (SGA), large for gestational age (LGA), stillbirth, low Apgar score (<7) at 5 min, and low Apgar score in term babies (Apgar < 7 and ≥37 weeks of gestation). SGA and LGA were defined as weights below the 10^th^ or above the 90^th^ percentiles for age, respectively, according to Australian reference values [25]. The three growth measurements were birthweight, birth length, and head circumference in term babies.

#### 2.2.2. Exposure and Other Variables

Mothers can move between exposure and comparison areas between pregnancies; therefore, the unit of analysis in this study was the pregnancy. We classified pregnancies as exposed or comparison if the mother’s recorded address at the time of birth in the PDC was in an exposure or comparison area, respectively. All covariates (see below) were sourced from the PDCs.

### 2.3. Statistical Analysis

All outcomes were analysed separately by exposure area. We used modified Poisson regression models with robust estimation of error variance to estimate relative risks (RR) and 95% confidence intervals (CI) for binary outcomes [26]. We used linear regression models to estimate the difference in means between groups and 95% CIs for growth measurements.

We decided *a priori* not to pool study results across exposure areas as environmental risk assessments indicated that the nature and sources of exposure (e.g., contaminated groundwater vs. contaminated municipal water) were different across the three exposure areas over the study period.

We specified two models for each outcome. In the first model, we adjusted for maternal age, maternal Aboriginal and Torres Strait Islander status (Indigenous, non-Indigenous), and year of birth; gestational age (37, 38, 39, 40, ≥41 weeks) was included for outcomes restricted to term babies. We refer to the above as a ‘minimally-adjusted model’.

In the second model, which we refer to as a ‘fully-adjusted model’ we additionally adjusted for the following where relevant: maternal country of birth (Australia, overseas), parity (0, 1, ≥2), marital status (married or de facto, other), pre-pregnancy body mass index (BMI in kg/m^2^), any smoking during pregnancy (smoker, non-smoker), baby sex (male, female), and macrosomia (birthweight ≥ 4000 g) (the specific adjustments for each outcome are described in the table of results). Year of birth, maternal age, and pre-pregnancy BMI were treated as continuous variables, all of which were modelled as natural cubic splines with three knots. Some covariates were not collected in all years; thus, fully adjusted models generally contained fewer years of data than minimal models.

We used a generalised estimating equations method with exchangeable correlation structures to account for repeated measures (mothers who had more than one pregnancy over the study period). In sensitivity analyses, we (1) included gestational diabetes as an additional covariate for the following outcomes: caesarean or assisted vaginal birth, LGA, and term birthweight; and (2) treated the following covariates as categorical variables: year of birth (10-year bands for the NT and NSW, 5-year bands for Qld), maternal age (0–19, 20–24, 25–29, 30–34, ≥35 years), and pre-pregnancy BMI (0 -< 18.5, 18.5 -< 25, 25 -< 30, ≥30 kg/m^2^).

As a supplementary analysis, we analysed a composite adverse infant outcome comprising any of the following events: preterm birth, SGA, LGA, stillbirth, and low Apgar score.

All data analyses and graphs for this report were generated using SAS software (version 9.4).

## 3. Results

### 3.1. Description of the Study Population

Our largest sample (of singleton pregnancies) was in the NT (5606 (33%) exposed, 11,364 (67%) comparison), followed by Qld (665 (14%) exposed, 3989 (86%) comparison), and NSW (188 (2.5%) exposed, 7287 (97.5%) comparison).

Sample sizes and characteristics by state and exposure status can be seen in Table 1. Each pair of exposed and comparison populations was similar in terms of maternal country of birth, maternal age, marital status, and Aboriginal and Torres Strait Islander status (apart from a higher proportion of mothers of Aboriginal and Torres Strait Islander descent in Oakey, relative to its comparison population). Based on Qld data, the exposed and comparison populations were reasonably well-matched on socioeconomic variables (75% and 99% of the comparison population lived in areas within the same Index of Relative Socioeconomic Disadvantage (IRSD) decile and remoteness category, respectively, as its exposed population) (Table 1).

### 3.2. Perinatal Outcomes in Relation to Living in Exposure Areas

The proportions of cases, adjusted RRs, mean growth measurements, and adjusted differences in means are shown in Table 2; a forest plot of adjusted RRs is shown in Figure 1.

In the NT, after adjusting for sociodemographic characteristics and other potential confounders, point estimates for all binary outcomes were not large, and all interval estimates were compatible with no differences in risks between Katherine and its comparison areas. In terms of growth measurements, term babies born to mothers who had lived in Katherine were larger than those in its comparison areas on all outcomes after minimal adjustment. However, only birth length remained higher in Katherine after full adjustment, albeit only marginally.

In Qld, the adjusted risk of stillbirth in Oakey was 2.6 times that of its comparison areas, although there was considerable uncertainty due to a small number of cases (fully-adjusted RR = 2.59, 95% CI 1.25, 5.39). There was uncertain evidence of an increased risk of low Apgar score (fully-adjusted RR = 1.47, 95% CI 0.95, 2.26), and when restricted to term babies, the results for this outcome were too imprecise to make any conclusions (fully-adjusted RR = 1.21, 95% CI 0.69, 2.10). For the other binary outcomes, point estimates were small and interval estimates were compatible with no effect. There was little to no meaningful difference between Oakey and its comparison areas in any of the growth measurements.

In NSW, the adjusted risk of postpartum haemorrhage in Williamtown was almost twice that of its comparison areas (fully-adjusted RR = 1.94, 95% CI 1.13, 3.33). Note that postpartum haemorrhage data were collected in the NSW PDC only from 2016; therefore, estimates were based on a small number of events. The risk of pregnancy-induced hypertension in Williamtown was nearly twice that of its comparison areas (fully-adjusted RR = 1.88, 95% CI 1.30, 2.73). For the remaining binary outcomes examined, interval estimates were compatible with no effect. While some point estimates were modest in size (for example, spontaneous preterm birth and term low Apgar score), interval estimates were too imprecise to conclude that rates were likely to be different. We found no evidence of meaningful differences in birthweight between Williamtown and its comparison areas.

In sensitivity analyses (Appendix A), there were no appreciable changes in effect sizes or direction of findings when we treated year of birth, maternal age and pre-pregnancy BMI as categorical instead of continuous variables. Including gestational diabetes as an additional covariate resulted in a reduction in the adjusted RR for LGA in the NT from 0.91 (95% CI 0.80, 1.04) to 0.86 (95% CI 0.74, 0.99) (Appendix A). In a supplementary analysis of a composite adverse infant outcome variable, we did not see evidence of increased risks in the exposure areas compared to their comparison areas (Appendix A).

## 4. Discussion

We estimated increased risks of some adverse perinatal outcomes in mothers who gave birth while living in Oakey and Williamtown. For other outcomes examined in these two areas and for all outcomes in Katherine, we could not conclude that the risks were different from their respective comparison populations. We did not find meaningful differences in growth measurements.

The findings for pregnancy-induced hypertension are of particular interest as previous studies have reported positive associations. In communities of highly exposed mothers from the mid-Ohio Valley region, some studies have suggested weak to moderate associations between maternal serum PFOA/PFOS and preeclampsia [27,28] or pregnancy-induced hypertension [29], while one study did not find an association after incorporating lifetime residential history and environmental/pharmacokinetic modelling of PFOA exposure [30]. In pregnancy cohorts with background levels of exposure, there has largely been no evidence of associations between various PFAS measured in blood and gestational hypertension or preeclampsia [31,32,33]. However, a recent study observed an association with late-onset preeclampsia and noted that the inconsistencies in previous findings may be due in part to not considering preeclampsia subtypes [34].

The elevated risks of pregnancy-induced hypertension and postpartum haemorrhage in Williamtown were accompanied by moderately elevated point estimates for preterm birth and gestational diabetes (but with confidence intervals that included the null). Gestational diabetes increases the risk of pregnancy-induced hypertension [35], and preterm birth rates are higher in mothers with diabetes or hypertension or both [36]. Pregnancy-induced hypertension is also a risk factor for postpartum haemorrhage [37], as is obesity [38], which is also related to gestational diabetes. As such, it is possible that the increased risks observed in these adverse outcomes were confounded by BMI. BMI was not collected in NSW until 2016, hence the sample size was too small for a BMI-adjusted analysis.

The largest relative effect estimated was the 2.6-fold risk of stillbirth in Oakey. The current evidence on the link between PFAS and stillbirth is sparse. Two studies from the C8 Health Project in communities exposed to PFOA did not report associations with stillbirth based on approximately 100 cases [27,30]. We did not find the risk of stillbirth to be elevated in Katherine and had limited power to detect an effect in Williamtown. Given a lack of prior evidence and the large number of analyses conducted, the positive association seen in Oakey should be viewed cautiously. In addition, we cannot rule out that Oakey mothers had a higher risk of stillbirth due to differences in factors unrelated to PFAS, including antenatal care [39] that we could not control for, which may warrant further attention.

In terms of growth measurements, meta-analyses have supported a relationship between maternal serum PFOA/PFOS and small reductions in birthweight [40,41,42]. However, some recent analyses have suggested that these findings depend on the timing of blood sampling, and there was little or no association when serum PFOS or PFOA were measured at the beginning of pregnancy [43,44,45]. We did not find any associations with birthweight or head circumference after full adjustment but saw minimal elevation of birth length (≤0.3 cm) in two exposure areas, which were in opposite directions to the hypothesised reduced growth associated with exposure. Studies that measured fetal growth by ultrasound have not reported associations with PFAS exposure [46,47]. Either way, small differences in growth measurements within normal limits are doubtful indicators of adverse impacts on health.

PFAS have been shown to cross the placenta [14,15] and are hypothesised to disrupt placental growth and function, thereby increasing the risk of adverse perinatal outcomes [48]. Placental dysfunction has been linked to hypertensive disorders of pregnancy [49] and low birthweight [50]. However, specific mechanisms for PFAS-induced placental damage as a driver of adverse perinatal outcomes have not been validated experimentally.

The strength of this study was the unbiased selection of all mothers who ever lived in the exposure areas at the time of giving birth since the inception of jurisdictional perinatal data collections. However, we were still limited by small numbers given the relatively few births in Oakey and Williamtown over this time, and we were not able to capture mothers who were exposed prior to data collection (PFAS exposure in Australia is possible as early as the 1970s).

Furthermore, we did not have information on mothers’ residential history (and thus her exposure over time) but only on her residence at the time of birth. Savitz and colleagues reported a moderate correlation (Spearman rank order correlation = 0.64) between estimated exposure based on lifetime residential history and maternal residence recorded on the baby’s birth certificate, but this is likely context-specific [27].

We must be cautious about drawing causal implications from this study on the effects of PFAS on perinatal outcomes. Our use of an ecological measurement of exposure means that individual-level exposure is inaccurate, and we cannot be sure that those with adverse outcomes had higher PFAS exposure than those without such outcomes (or vice versa).

However, our use of an ecological measurement of exposure (e.g., place of residence) avoided confounding and the need to control for such confounding that can occur when using personal measurements of exposure [51]. This includes physiological or behavioural factors that affect personal exposure, such as those that affect PFAS absorption in the body. For example, it has been suggested that the association between maternal or cord serum PFAS and birthweight may be confounded by maternal factors that affect both the glomerular filtration rate (thus PFAS elimination) and birthweight [45]. 

Other advantages included being able to involve a larger sample size and historical populations, which would not have been possible in studies measuring individual exposure and other potentially confounding individual factors. However, the trade-off of this approach is higher exposure measurement error, with a potential bias towards the null.

We also could not account for behavioural risk factors such as alcohol and diet, and there were insufficient years of data collected for BMI and marital status in the NT and NSW to allow adjustments for these variables. We did not have information on fathers’ occupational exposure to chemicals, which may affect male reproductive health and is an active area of research [52].

We did not have individual measures of socioeconomic status in the NT and NSW to assess the selected comparison populations; however, based on Qld data, Oakey and its comparison population appear reasonably well-matched on IRSD and remoteness categories (see Table 1). Finally, the scope of the PDCs includes live births and stillbirths of at least 20 weeks’ gestation. Therefore, any potential relationships between PFAS and earlier outcomes, such as infertility or miscarriage [53], would not have been detected in this study.

## 5. Conclusions

Our study found limited evidence for increased risks of adverse perinatal outcomes in mothers living in two of three Australian exposure areas with PFAS contamination from firefighting foams. Our finding on pregnancy-induced hypertension was somewhat consistent with evidence from the C8 Health Study. However, our other positive findings are not supported by prior evidence; none were consistent across exposure areas, and we could not reasonably rule out chance, bias or residual confounding.

## Figures and Tables

**Figure 1 ijerph-20-06886-f001:**
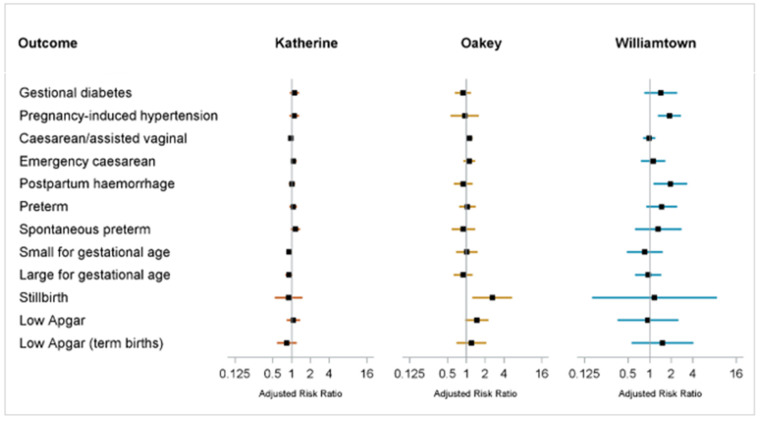
Forest plot showing Model 2 adjusted relative risks (RR) for adverse perinatal outcomes. Data sources: NSW Perinatal Data Collection (1994–2018), NT Perinatal Trends (1986–2017), Qld Perinatal Data Collection (2007–2018); Forest plot shows point estimates of adjusted RRs (filled squares) from Model 2 and associated 95% confidence interval (horizontal lines) and solid vertical line of no effect; Model 2 RRs were adjusted for year of birth, maternal age, maternal Aboriginal and Torres Strait Islander status (except NSW), parity, marital status (except NSW), maternal country of birth, maternal BMI (Qld only), and maternal ever smoked during pregnancy. Caesarean/assisted vaginal, emergency caesarean and postpartum haemorrhage were additionally adjusted for macrosomia. Preterm birth, stillbirth, low Apgar and growth measures were additionally adjusted for sex. Outcomes restricted to term babies included adjustment for gestational week. See Table 2 for sample sizes, crude risks and adjusted RRs. Adjusted RRs are on a log scale.

**Table 1 ijerph-20-06886-t001:** Characteristics of the study populations, the NT (1986–2017), Qld (2007–2018), NSW (1994–2018).

	NT		Qld		NSW	
Characteristic	Exposed n (%)	Comparison n (%)	Exposed n (%)	Comparison n (%)	Exposed n (%)	Comparison n (%)
Mothers						
Total sample ^1^	4083	8607	513	2876	144	4871
Country of birth						
Australia	3529 (87)	6825 (80)	463 (90)	2642 (92)	130 (90)	4522 (93)
Overseas	543 (13)	1720 (20)	50 (10)	234 (8)	14 (10)	349 (7)
Missing/unknown	11	62	0	0	0	0
Indigenous status ^2^						
No	2916 (71)	6795 (79)	424 (83)	2707 (94)	139 (97)	4651 (96)
Yes	1167 (29)	1806 (21)	89 (17)	169 (6)	≤5	206 (4)
Missing/unknown	0	6	0	0	≤5	14
Pregnancies						
Total sample ^1^	5606	11,364	665	3989	188	7287
Year of birth						
1985–1994	1731 (31)	2457 (22)	N/A	N/A	≤5	264 (4)
1995–2004	1724 (31)	3466 (31)	N/A	N/A	41 (22)	2754 (38)
2005–2014	1736 (31)	4157 (37)	445 (67)	2681 (67)	100 (53)	3033 (42)
2015–latest	415 (7)	1283 (11)	220 (33)	1308 (33)	44 (23)	1236 (17)
Missing/unknown	0	1	0	0	≤5	0
Mother’s age (at baby’s birth) ^3^						
<20	606 (11)	925 (8)	87 (13)	360 (9)	≤5	332 (5)
20–24	1344 (24)	2182 (19)	201 (30)	961 (24)	22 (12)	1284 (18)
25–29	1714 (31)	3187 (28)	194 (29)	1281 (32)	60 (32)	2374 (33)
30–34	1315 (23)	3252 (29)	121 (18)	865 (22)	60 (32)	2105 (29)
35–39	526 (9)	1534 (14)	62 (9)	522 (13)	39 (21)	976 (13)
40+	101 (2)	281 (2)	N/A	N/A	6 (3)	215 (3)
Missing/unknown	0	3	0	0	≤5	1
Gestational age (weeks)						
≤36	455 (8)	929 (8)	61 (9)	314 (8)	15 (8)	399 (5)
37–40	4298 (77)	9046 (80)	524 (79)	3209 (80)	139 (74)	5553 (76)
41+	827 (15)	1300 (12)	80 (12)	465 (12)	34 (18)	1332 (18)
Missing	26	89	0	1	0	3
Baby sex						
Female	2688 (48)	5607 (49)	329 (49)	1899 (48)	94 (50)	3552 (49)
Male	2916 (52)	5753 (51)	336 (51)	2090 (52)	94 (50)	3735 (51)
Missing/unknown	2	4	0	0	0	0
Maternal parity						
No prior birth	2087 (37)	4683 (41)	190 (29)	1033 (26)	72 (38)	2595 (36)
One prior birth	1676 (30)	3501 (31)	160 (24)	1054 (26)	69 (37)	2436 (33)
≥Two prior births	1840 (33)	3156 (28)	315 (47)	1902 (48)	47 (25)	2246 (31)
Missing/unknown	3	24	0	0	0	10
Marital status (at birth) ⁴						
Married/de facto	3610 (65)	7156 (64)	484 (73)	3133 (79)	12 (80)	960 (87)
Other	1959 (35)	4014 (36)	181 (27)	856 (21)	3 (20)	148 (13)
Missing/unknown	37	194	0	0	173	6179
Pre-pregnancy BMI ⁵						
0 -< 18.5	13 (3)	48 (3)	25 (4)	210 (5)	≤5	34 (4)
18.5 -< 25	209 (43)	772 (53)	253 (41)	1596 (41)	12 (41)	404 (44)
25 -< 30	139 (29)	393 (27)	160 (26)	1012 (26)	≤10	244 (27)
≥30	125 (26)	257 (17)	180 (29)	1072 (28)	9 (31)	228 (25)
Missing/unknown	5120	9894	47	99	159	6377
Smoking during pregnancy ⁶						
No	2293 (68)	5623 (75)	471 (71)	2959 (74)	158 (84)	5909 (81)
Yes	1092 (32)	1912 (25)	191 (29)	1025 (26)	30 (16)	1355 (19)
Missing/unknown	2221	3829	3	5	0	23
IRSD ⁷ decile						
1	N/A	N/A	0	97 (2)	N/A	N/A
2	N/A	N/A	0	599 (15)	N/A	N/A
3	N/A	N/A	665 (100)	2966 (74)	N/A	N/A
4	N/A	N/A	0	327 (8)	N/A	N/A
5–10	N/A	N/A	0	0	N/A	N/A
Remoteness						
Major city	N/A	N/A	0	0	N/A	N/A
Inner regional	N/A	N/A	0	0	N/A	N/A
Outer regional	N/A	N/A	0	43 (1)	N/A	N/A
Remote	N/A	N/A	665 (100)	3946 (99)	N/A	N/A
Very remote	N/A	N/A	0	0	N/A	N/A

N/A: not applicable; Denominators for proportions exclude missing values; Cells have been suppressed to avoid reporting cell numbers with size ≤ 5; Percentages were rounded to integer values. Data sources: NSW Perinatal Data Collection (1994–2018), the NT Perinatal Data Collection (1986–2017), Qld Perinatal Data Collection (2007–2018). ^1^ Mothers can move between exposure and comparison areas; therefore, totals are not for unique mothers. ^2^ Mothers who had two or fewer perinatal records were coded as Aboriginal and Torres Strait Islander if the mother identified as Aboriginal and/or Torres Strait Islander at least once. Mothers who had more than two pregnancies were coded as Aboriginal and Torres Strait Islander if the mother identified as Aboriginal and/or Torres Strait Islander at least two times. ^3^ Mother’s age is top-coded at 35 years in the Qld Perinatal Data Collection (PDC). ^4^ Marital status is only available from 1994–1997 in the NSW PDC and thus was not used as a covariate due to insufficient data. ^5^ BMI is only available from 2014–2017 in the NT PDC and 2016–2018 in the NSW PDC and thus was not used as a covariate in these states due to insufficient data. ^6^ Smoking during pregnancy is only available from 1996 in the NT PDC. ^7^ Index of Relative Socioeconomic Disadvantage (IRSD) based on Statistical Area Level 2 of the mother’s usual address coded according to the Australian Bureau of Statistics Australian Statistical Geography Standard (ASGS) 2011 Version for births up to 2016/2017 and ASGS 2016 Version for births from 2017/2018. IRSD decile and remoteness area were not available in the NT and QLD PDCs.

**Table 2 ijerph-20-06886-t002:** Comparison of perinatal outcomes in the exposed and comparison populations: risks (%) and adjusted relative risks (RR) of adverse perinatal outcomes, and means and adjusted difference in means of growth measurements.

	NT				Qld				NSW			
	Exposed % (n)	Comparison % (n)	Adjusted RR ^1^ (95% CI)	Adjusted RR ^2^ (95% CI)	Exposed % (n)	Comparison % (n)	Adjusted RR ^1^ (95% CI)	Adjusted RR ^2^ (95% CI)	Exposed % (n)	Comparison % (n)	Adjusted RR ^1^ (95% CI)	Adjusted RR ^2^ (95% CI)
Adverse Perinatal Outcome												
Total sample	5606	11,364			665	3989			188	7287		
Gestational diabetes ^3^	7% (215)	7% (505)	1.08 (0.92,1.28)	1.11 (0.94,1.32)	8% (53)	9% (361)	0.95 (0.71,1.27)	0.88 (0.66,1.19)	8% (13)	5% (306)	1.44 (0.85,2.44)	1.42 (0.84,2.41)
Pregnancy-induced hypertension	5% (294)	5% (607)	0.93 (0.80,1.07)	1.10 (0.92,1.32)	3% (17)	2% (99)	1.03 (0.62,1.72)	0.94 (0.56,1.59)	16% (30)	7% (527)	2.00 (1.36,2.93)	1.88 (1.30,2.73)
Caesarean/assisted vaginal	31% (1711)	34% (3853)	0.99 (0.95,1.05)	0.97 (0.91,1.03)	40% (263)	36% (1427)	1.13 (1.01,1.27)	1.12 (1.00,1.25)	38% (71)	33% (2423)	1.03 (0.84,1.26)	0.98 (0.81,1.20)
Emergency caesarean	11% (607)	12% (1306)	1.00 (0.90,1.10)	1.08 (0.96,1.21)	14% (90)	12% (490)	1.10 (0.88,1.37)	1.11 (0.89,1.39)	13% (25)	10% (747)	1.22 (0.83,1.81)	1.12 (0.76,1.65)
Postpartum haemorrhage ⁴	8% (475)	9% (1075)	0.96 (0.86,1.06)	1.01 (0.90,1.14)	6% (43)	7% (265)	0.96 (0.70,1.32)	0.89 (0.63,1.26)	34% (10)	19% (174)	1.97 (1.14,3.38)	1.94 (1.13,3.33)
Preterm birth	8% (455)	8% (929)	0.95 (0.85,1.07)	1.06 (0.92,1.22)	9% (61)	8% (314)	1.13 (0.85,1.49)	1.04 (0.77,1.41)	8% (15)	5% (399)	1.46 (0.89,2.39)	1.47 (0.90,2.40)
Spontaneous preterm birth	6% (310)	5% (586)	0.99 (0.86,1.14)	1.14 (0.96,1.36)	4% (29)	5% (180)	0.97 (0.65,1.44)	0.90 (0.58,1.39)	4% (7)	3% (234)	1.27 (0.60,2.68)	1.32 (0.63,2.77)
Small for gestational age (SGA)	13% (750)	14% (1553)	0.87 (0.79,0.94)	0.92 (0.82,1.03)	5% (30)	5% (192)	0.93 (0.63,1.37)	1.02 (0.69,1.51)	7% (13)	9% (621)	0.87 (0.48,1.57)	0.86 (0.48,1.53)
Large for gestational age (LGA)	8% (453)	9% (1028)	0.96 (0.86,1.08)	0.91 (0.80,1.04)	6% (39)	6% (259)	0.93 (0.66,1.32)	0.89 (0.63,1.27)	11% (20)	11% (836)	0.89 (0.58,1.37)	0.95 (0.62,1.45)
Stillbirth	1% (45)	1% (89)	0.94 (0.66,1.35)	0.90 (0.54,1.50)	2% (11)	1% (27)	2.43 (1.20,4.93)	2.59 (1.25,5.39)	(≤5)	<1% (34)	1.13 (0.15,8.29)	1.17 (0.16,8.61)
Low Apgar score at 5 min	3% (164)	3% (327)	0.93 (0.77,1.13)	1.07 (0.83,1.37)	4% (29)	3% (113)	1.50 (0.99,2.26)	1.47 (0.95,2.26)	(≤5)	2% (161)	0.98 (0.37,2.57)	0.95 (0.36,2.51)
Term (≥37 weeks) outcome												
Total sample	5125	10,346			604	3674			173	6885		
Term low Apgar score at 5 min	2% (80)	2% (185)	0.77 (0.59,1.00)	0.84 (0.59,1.20)	2% (14)	2% (70)	1.15 (0.66,2.00)	1.21 (0.69,2.10)	(≤5)	2% (103)	1.57 (0.58,4.20)	1.51 (0.56,4.07)
	Exposed mean (SD)	Comparison mean (SD)	Adjusted difference in means ^1^ (95% CI)	Adjusted difference in means ^2^ (95% CI)	Exposed mean (SD)	Comparison mean (SD)	Adjusted difference in means ^1^ (95% CI)	Adjusted difference in means ^2^ (95% CI)	Exposed mean (SD)	Comparison mean (SD)	Adjusted difference in means ^1^ (95% CI)	Adjusted difference in means ^2^ (95% CI)
Growth measure												
Total sample	5125	10,346			604	3674			173	6885		
Term birth weight (g)	3423 (488.9)	3405 (488.9)	31.1 (14.4,47.9)	11.22 (−8.9,31.3)	3465 (448.8)	3483 (468.9)	−9.6 (−49.7,30.6)	−11.1 (−48.7,26.5)	3519 (479.6)	3515 (477.4)	21.7 (−49.0,92.1)	35.7 (−29.4,100.8)
Term birth length (cm) ⁵	50.7 (2.7)	50.3 (2.4)	0.3 (0.1,0.4)	0.2 (0.1,0.4)	51.6 (2.6)	51.4 (2.6)	0.3 (0.1,0.5)	0.3 (0.1,0.5)	Data not available			
Term head circumference (cm) ⁶	34.7 (1.7)	34.6 (1.5)	0.1 (0.0,0.2)	0.0 (−0.1,0.1)	34.7 (1.6)	34.7 (1.5)	0.0 (−0.1,0.2)	0.0 (−0.1,0.2)				

Notes: The RR is the risk in the exposed group divided by the risk in the comparison group. The difference in means is the mean in the exposed group minus the mean in the comparison group. ^1^ RRs/Difference in means from Model 1: adjusted for year of birth, maternal age, and mother’s Aboriginal and Torres Strait Islander status (except NSW). Outcomes restricted to term babies included adjustment for gestational week. ^2^ RRs/Difference in means from Model 2: adjusted for year of birth, maternal age, maternal Aboriginal and Torres Strait Islander status (except NSW), parity, marital status (except NSW), maternal country of birth, maternal BMI (Qld only), and maternal ever smoked during pregnancy. Caesarean/assisted vaginal, emergency caesarean and postpartum haemorrhage were additionally adjusted for macrosomia. Preterm birth, stillbirth, low Apgar, and growth measures were additionally adjusted for the sex of the baby. Outcomes restricted to term babies included adjustment for gestational week. RRs from Model 2 are represented in a forest plot in Figure 1. ^3^ Gestational diabetes is only available in the NSW Perinatal Data Collection (PDC) from 1994 to 2015; the denominators for exposed and comparison are 159 and 6345, respectively. Gestational diabetes is only available in the NT PDC from 2000; the denominators for exposed and comparison are 2943 and 7229, respectively. ^4^ Postpartum haemorrhage is only available in the NSW PDC from 2016; the denominators for exposed and comparison are 29 and 938, respectively. For this outcome, year was modelled as a categorical rather than continuous covariate. ^5^ Birth length is only available in the NT PDC from 2008; the denominators for exposed and comparison are 1422 and 3024, respectively. ^6^ Head circumference is only available in the NT PDC from 2008; the denominators for exposed and comparison are 1391 and 2954, respectively.

## Data Availability

The datasets supporting the conclusions of this article are stored on password-protected local area networks and are not publicly available under data disclosure agreements. The authors acknowledge the NSW Ministry of Health, the NT Department of Health, Queensland Health, the Centre for Health Record Linkage and Data Linkage Queensland for facilitating access to and linkage of data.

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
