# Peer review of "Relative Risks of Adverse Perinatal Outcomes in Three Australian Communities Exposed to Per- and Polyfluoroalkyl Substances: Data Linkage Study"

_ijerph, 2023, doi:10.3390/ijerph20196886_

Round 1
Reviewer 1 Report
Dear authors congratulations
what a nice paper
to evaluate the impact of fire fight foams on pregnancy outcome is absolutely of interest nowday not only in australia but also in all mediterranean countries
i'm particularlu courious to ask to better define and describe the Qld population which has had a higher rate of stillbirth
please better discuss this population charachteristics within the discussion and mention how crucial is in general to detect small for gestational age and obstetric complications such as GDM in the prevention of stillbirth please read and cite
doi: 10.36129/jog.2022.20
best regards
Author Response
Thank you for your comments.
In the fourth paragraph of the Discussion, we have added: In addition, we cannot rule out that Oakey mothers had a higher risk of stillbirth due to differences in factors unrelated to PFAS including antenatal care that we could not control for, which may warrant further attention. (ref Quaresima P, Saccone G, Morelli M, Interlandi F, Votino C, Zuccalà V, Di Carlo C, Zullo F, Venturella R. Stillbirth, potentially preventable cases: An Italian retrospective study. Ital. J. Gynaecol. Obstet. 2022;34:89-102.)
Reviewer 2 Report
The authors addressed important issue of the sequels of PFAS exposure in terms of adverse perinatal outcomes. Analysis has been performed on a well-defined and meticulously identified populations of 3 Australian areas with known exposures to PFAS. Assessed samples of pregnant women from respective areas are sufficiently numerous to allow credible analysis and comparison groups have also been properly recruited from the same areas.
Data are presented in a clear, logical and convincing way. There isn’t anything to reproach from the point of view of the selection of statistical methods and carrying out the analysis, either.
Obtained results are critically discussed with provisions made for possibilities of other ways of sample selection which could influence the outcomes. The most important in this context is the inaccessibility of mothers’ residential history which can contribute to intersubject difference in exposure and thus impact the analysis results. However, the authors critically address this possibility and provide explanation presenting advantages and drawbacks of their approach.
One thing that looks missing to me in the whole approach are babies’ fathers-related factors. Are there any data available on e.g., fathers’ age at conception, concomitant diseases or occupational exposure to chemicals. There are data regarding possible influence of exercising a firefighter profession upon male fertility. Also, FPAS and other endocrine disrupting chemicals contained in firefighting foams can influence fertility – male data are more predominant in this aspect, with considerable knowledge gap existing with regard to female population.
I realize that accession to data regarding male partners of mothers whose pregnancy and labor-related events had been analyzed, yet the authors may consider including a short passage referring to this aspect in either introduction or discussion.
Otherwise there is not much to amend in the manuscript.
Please find below the suggestion for minor typo issue.
ABSTRACT: Conclusions section should start with „There was…” instead of „here was…”
Author Response
Thank you for your comments.
In the second last paragraph of the Discussion, we have added: We did not have information on fathers’ occupational exposure to chemicals, which may affect male reproductive health and is an active area of research. (ref Bach CC, Vested A, Jorgensen KT, Bonde JPE, Henriksen TB, Toft G. Perfluoroalkyl and polyfluoroalkyl substances and measures of human fertility: a systematic review, Critical Reviews in Toxicology 2016; 46(9), 735-755; DOI: 10.1080/10408444.2016.1182117)
We have edited the error in the abstract.